# Interrupted Time Series Analysis of Pediatric Infectious Diseases and the Consumption of Antibiotics in an Atlantic European Region during the SARS-CoV-2 Pandemic

**DOI:** 10.3390/antibiotics11020264

**Published:** 2022-02-18

**Authors:** Ana Clavería, María Victoria Delgado-Martín, Ana Goicoechea-Castaño, José Manuel Iglesias-Moreno, Clara García-Cendón, María Victoria Martín-Miguel, Rita Villarino-Moure, Carolina Barreiro-Arceiz, Isabel Rey-Gómez-Serranillos, Javier Roca

**Affiliations:** 1Galicia South Health Research Institute, Health Area of Vigo, Galician Health Service, 36201 Vigo, Spain; anaclaveriaf@gmail.com; 2Network for Research on Chronicity, Primary Care, and Health Promotion (RICAPPS), 08007 Barcelona, Spain; roca@uvigo.es; 3Moaña Health Center, Health Area of Vigo, Galician Health Service, 36950 Vigo, Spain; marividelgadomartin@gmail.com; 4Sárdoma Health Center, Health Area of Vigo, Galician Health Service, 36214 Vigo, Spain; ana.goicoechea.castano@sergas.es (A.G.-C.); rita.villarino.moure@sergas.es (R.V.-M.); 5Val Miñor Health Center, Health Area of Vigo, Galician Health Service, 36350 Vigo, Spain; josemanuel.iglesias.moreno@sergas.es (J.M.I.-M.); claracendon75@gmail.com (C.G.-C.); 6Vigo Family and Community Medicine and Nursing Teaching Unit, Health Area of Vigo, Galician Health Service, 36201 Vigo, Spain; 7Baiona Health Center, Health Area of Vigo, Galician Health Service, 36201 Vigo, Spain; carobarreiroarceiz@gmail.com; 8Pharmaceutical Services, Health Area of Vigo, Galician Health Service, 36201 Vigo, Spain; isabel.rey.gomez.serranillos@sergas.es; 9Centro de Investigación e Tecnoloxía Matemática de Galicia (CITMAGA), Campus Vida, 15782 Santiago de Compostela, Spain; 10Department of Statistics and Operational Research, University of Vigo, 36310 Vigo, Spain

**Keywords:** interrupted time series analysis, pediatric, infectious diseases, antibacterial agents, SARS-CoV-2, drug utilization, Spain

## Abstract

The increasing concern about bacterial resistance has made the rational prescription of antibiotics even more urgent. The non-pharmacological measures established to reduce the impact of the SARS-CoV-2 pandemic have modified the epidemiology of pediatric infections and, consequently, the use of antibiotics. Interrupted time series (ITS) analyses are quasi-experimental studies that allow for the estimation of causal effects with observational data in “natural experiments”, such as changes in health policies or pandemics. The effect of the SARS-CoV-2 pandemic on the incidence of infectious diseases and the use of antibiotics between 2018 and 2020 in the Health Area of Vigo (Galicia, Spain) was quantified and analyzed. This paper outlines a real-world data study with administrative records from primary care services provided for the pediatric population. The records were related to episodes classified as infectious by the International Classification of Primary Care (ICPC-2) and oral medication in the therapeutic subgroup J01, corresponding to antibiotics for systemic use, according to the World Health Organization’s Anatomical Therapeutic Chemical (ATC) classification system. The records were classified according to incident episodes, age, dose per inhabitant, and year. Segmented regression models were applied using an algorithm that automatically identifies the number and position of the change points. During the SARS-CoV-2 pandemic, the number of infectious diseases being transmitted between individuals, through the air and through the fecal–oral route, significantly decreased, and a slight decrease in infections transmitted via other mechanisms (urinary tract infections) was also found. In parallel, during the months of the pandemic, there has been a marked and significant reduction in antibacterial agent utilization, mainly of penicillins, cephalosporins, and macrolides.

## 1. Introduction

Since their discovery by Fleming [1] until the present day, the use of antibiotics has undergone considerable changes. The excessive prescription over recent decades, including that of broad-spectrum antibiotics, has led to the appearance of antimicrobial resistance [2,3,4,5,6]. Therefore, in 2014, the Spanish government approved the National Plan Against Resistance to Antibiotics (PRAN) [7]. According to data from 2020, bacterial isolate resistance in Spain was 4.7% [8], higher than that of countries such as Sweden, France, or Germany, with rates of 0.3%, 0.5%, and 0.5%, respectively. Other countries such as Greece or Bulgaria were found to have the highest rates, at 66% and 28.1%, respectively. Fortunately, an increased awareness regarding the rational use of antibiotics has been promoted by healthcare workers, although we are far from achieving global awareness of this issue.

In the childhood population, following clinical protocols and guidelines has helped to ensure the appropriate use of antibiotics, reverting the trend in Spain and making antibiotic use in pediatric-aged patients comparable to that of the rest of Europe [9,10]. Childhood infections tend to be viral in nature (upper respiratory tract and gastrointestinal infections, etc.), with bacterial infections being less common (although they do exist, and some of them are extremely serious). Bacterial infections occurring in children include respiratory tract infections (pharyngoamygdalitis, acute otitis media, and pneumonia). Urinary tract infections also affect children, with a prevalence of 3 to 7% [11]. They occur in 8–10% of all girls and in 2–3% of boys prior to the age of 7, being most common in males during the first three months of life and becoming increasingly common in girls after one year of age [12].

The global pandemic caused by SARS-CoV-2 has been a major turning point in healthcare and the social organization of many countries, including Spain. The declaration of the country’s first “state of alarm” in response to SARS-CoV-2 on 14 March 2020 (Royal Decree 463/2020) [13] limited mobility and implemented a lockdown of the Spanish population, and also led to the reorganization of the healthcare system to adapt to the emergency health situation.

In Galicia, like the rest of Spain, extraordinary measures were implemented in primary care, with telephone consultations being prioritized, a differentiated circuit for patients susceptible to SARS-CoV-2 being implemented, and telemedicine being used to assist them at home [14]. In May of 2020, a de-escalation process was initiated in the country, which ended on 16 September 2021, promoted by the onset of vaccination [15] and the availability of rapid antigen tests [16].

In the school environment, “bubble classrooms” were created, where the distance between desks had to be as far as possible. In addition, any activity that entailed the removal of face masks, such as eating or drinking, was prohibited, and each student’s footwear had to be replaced at the entrance of the classroom. Frequent hand hygiene, as well as maintaining cleanliness in the classroom and continuous ventilation, constituted a fundamental pillar for the enforced measures. The entering and exiting of centers was conducted in a staggered manner, maintaining an interpersonal distance of 1.5 m and restricting access to third parties.

During the pandemic, as a result of the decline in secondary respiratory pathogens resulting from the measures applied to reduce the transmission of SARS-CoV-2 (hand hygiene, the use of face masks, and social distancing), a change in the epidemiology of these infections took place. The incidence of non-COVID-19 respiratory viruses, including influenza, declined. In 2020, for the WHO European Region, non-COVID-19 respiratory viruses remained at baseline from the beginning of the season. In Spain, influenza activity remained lower than expected [17] as well. Decreases were also seen in bacterial infections. Distinct studies carried out in the primary care of distinct countries revealed an initial increase in the use of antibiotics at the onset of the pandemic, followed by a major decline, coinciding with the implementation of mobility restriction measures in an attempt to control the virus [18,19,20,21,22,23].

Interrupted time series (ITS) analyses are quasi-experimental studies permitting the estimation of causal effects for observational data. For this, regression models are used to assess “natural experiments”, such as changes in healthcare policies, the implementation of clinical guidelines, etc. Furthermore, their graphical and numeric representation can be easily understood, even by those who are not experts in statistics or epidemiology. Of the public health interventions analyzed using this type of methodology, there are studies on the use of medications, the introduction of legislation related to alcohol or tobacco, as well as the health impacts of catastrophic events, such as the economic or SARS-CoV-2 crises, quality improvement initiatives, among others published in recent reviews [24,25].

This article attempts to demonstrate the effects of the SARS-CoV-2 pandemic on the incidence of infectious illnesses in pediatric-aged patients and on the use of antibiotics. To do so, segmented regression models were applied using an algorithm that automatically identifies the number and position of the change points, permitting comparisons to be made with the study hypothesis.

## 2. Results

The pediatric population declined progressively from 74,431 in 2018 to 73,353 in 2019, and then to 72,483 in 2020 [26].

### 2.1. Number of Episodes by ICPC-2 Chapter

As for the number of infections by age group, in 2018, 2019, and 2020, a higher number of episodes was observed when the age group was lower, and a decrease was seen in the last year. Therefore, in children aged 0 to 4, the incidence went from 22,487 in 2018 to 23,044 in 2019, and then to 11,286 in 2020. For children aged 5 to 9, the number of episodes went from 9867, to 8725, to 7140, respectively. For the 10 to 14 age group, the evolution was from 6683, to 6060, to 5791. The disaggregated information by ICPC chapter, age group, and year is presented in Table 1.

As for the time pattern, incidence was calculated by dividing the number of diagnoses by the pediatric population at the half-way point of the period. Figure 1 presents the monthly evolution, adding respiratory and otorhinolaryngologic diagnoses to the diagnoses of Chapter A (which include primarily exanthematous viral diseases and SARS-CoV-2).

All the time series were adjusted with a first-order autocorrelation. Next, the number of minimum change points that significantly differentiate the regression models from the other methodology was identified. The resulting models for episodes of Chapter A, for respiratory and otorhinolaryngologic episodes, as well as the other episodes, are presented in Appendix B, including the confidence interval for the constants, slopes, significance, and the fit of the model.

The fit of the models was very good, with an adjusted R-squared of 0.9351, 0.9507, and 0.9614, respectively.

In Figure 1, the change point is represented by a broken red line, as are the confidence intervals of the series’ diverse regression lines. The periodicity of each series is shown, although for distinct periods of the year. As for the illnesses of Chapter A, they decreased at the start of 2020. However, a major change takes place in the constant after the summer, most likely explained by the start of the school year and the availability of SARS-CoV-2 tests. In the case of respiratory and otorhinolaryngologic diagnoses, the constant had a greater value at the onset of 2020, followed by a sharp decrease, coinciding with the lockdown.

### 2.2. Number of DHD by Therapeutic Subgroup

As for invoicing by therapeutic subgroup, Table 2 presents the annual evolution in 2018, 2019, and 2020. An annual decrease can be seen, which is more notable in the following groups: penicillins (J01C), cephalosporins (J01D), trimethoprim and derivatives (J01EA), and macrolides (J01F).

To perform the segmented regression, the monthly prescription filling value for the groups with the most invoicing (penicillins, cephalosporins, and macrolides (J01C, J01D, and J01F)) was used. The remaining active ingredients were added in the “Other antibiotics” category (J01A, J01E, J01M, J01X). The fitted models are presented in Appendix B (Table A1, Table A2, Table A3, Table A4, Table A5, Table A6 and Table A7).

Figure 2 shows the evolution of penicillins and macrolides, with five change points each. The changes were of a seasonal nature, with a large increase in the constant of the line after March 2020; a slight increase was seen in macrolides as compared to penicillins.

For the active ingredients added in the “Other antibiotics” category (J01A, J01E, J01M, J01X), the slopes are mostly horizontal (Figure 3).

The fit of the models was very good, with an adjusted R-squared of 0.946, 0.9284, and 0.9437, respectively.

## 3. Discussion

### 3.1. Main Results

The segmented regression methods revealed a significant effect of the SARS-CoV-2 pandemic on the infectious episodes registered in the primary care clinical histories of a pediatric population. In parallel, a significant decrease was evident in the prescription of antibiotics, especially penicillins, cephalosporins, and macrolides. As for the diseases of Chapter A, seven change points were observed, with a significant jump in September 2020 at the beginning of the school year when diagnostic tests for SARS-CoV-2 became available. Respiratory and otorhinolaryngologic episodes presented six change points, and the segmented regression corresponding with the period of confinement in the first months of 2020 shows a marked downward trend, followed by a more horizontal one.

Regarding the consumption of antibiotics (penicillins, cephalosporins, and macrolides), five significant change points were identified. The changes in constants and slopes for the identified segmented regressions followed a similar evolution to that observed in respiratory and otorhinolaryngologic diseases, with a very marked downward trend at the beginning of 2020, followed by a small change in the following months.

### 3.2. Impact on Infectious Diseases

In the pediatric population under study, a significant decrease was observed in respiratory infections starting in February of 2020 and reaching the largest decline between April and June of the same year (both included), coinciding with the implementation of harsher restriction measures to contain the SARS-CoV-2 pandemic (lockdown, school closing, and restricted mobility), and the resulting social isolation. In September of 2020, a slight increase was noted; this is likely to be related to the return to school activity. However, the levels remained considerably below those of 2018 and 2019.

Respiratory infections are the most common infections occurring during childhood, with children suffering from an average of three to eight upper respiratory infections per year, rates which may be twice as high in infants attending nursery school. Generally speaking, these infections tend to be caused by rhinoviruses, parainfluenza viruses, respiratory syncytial viruses, other coronaviruses, adenoviruses, enteroviruses, and influenza viruses [12]. The data collected here do not permit the analysis of the etiological agents in most of these infections. Therefore, we are unaware of the individual behavior of each of these viruses in our population. We can, however, infer that a global decline took place in these pathogens, given the major decrease observed in all types of respiratory infections, both upper (upper respiratory viruses, sinusitis, amygdalitis, laryngitis), as well as lower (bronchitis, pneumonia) for all age groups, as compared to the two prior years. One exception is influenza, which maintained a similar incidence as that of the two years prior to the period of study.

As for the other viral infections, the aggregate study of diagnoses from Chapter A of the ICPC (exanthematous viral diseases and SARS-CoV-2) reveals an increase in these illnesses in February of 2020, coinciding with the arrival and initial diagnoses of the SARS-CoV-2 virus in Spain. As of April 2020, this initial increase was followed by a progressive decline, which continued until June of the same year. As of this moment, a notable increase took place in these infections, which continued until the end of the studied period. This pattern is not normal for this group of illnesses, which tend to reveal two seasonal peaks, one in the spring and another in the fall, as observed in the data from 2018 and 2019.

As for the other infections examined, gastrointestinal and urinary tract infections are noteworthy. The former experienced a pronounced decrease as compared to the years prior to the pandemic, while the latter did not undergo any substantial changes, with only a minor decrease being noted in their incidence following the application of the restrictions.

These results are similar to those published by other authors, who have noted that the measures taken to contain the transmission of SARS-CoV-2 led to a global decline in the incidence of transmissible respiratory or contact infections, although they did not affect other infections that are unaffected by social contacts, such as urinary tract infections [27,28]. For these other infections, only a mild decline was observed, perhaps due to the lack of physician visits in response to mild and self-treatable symptoms.

The major decline in respiratory infections found in the study population has also been revealed by Haddadin [29], who noted intense drops in the incidence of acute respiratory infections, including those caused by the respiratory syncytial virus (RSV) and influenza. This latter infection has not been observed in our study. The incidence of influenza remained stable in 2020 as compared to the prior two seasons. This may be due to the fact that the pandemic in Spain took place in March of 2020, after the end of the annual flu epidemic, which took place during the months of January and February. WHO also notes that current influenza surveillance data should be interpreted with caution, as the ongoing SARS-CoV-2 pandemic has influenced the ability to seek medical assistance, affected staffing, and led to changes in priorities, as well as the ability to perform tests. This distortion continued at the beginning of the 2020–2021 flu season, forcing a rethinking and adaptation of influenza surveillance, following the international recommendations of the European Center for Disease Prevention and Control (ECDC) and the World Health Organization (WHO) [17].

Rius-Peris and Britton [30,31] reported a notable decline in admittance and assistance to urgent care due to bronchiolitis (caused by RSV) and a change in seasonality [29,30], which was also seen in other respiratory viruses [32]. Rhinovirus, however, which does not have a lipid coating, and therefore does not respond as well to hand hygiene for its prevention, continued to circulate atypically and was the agent responsible for most of the bronchiolitis cases in the 2020–2021 season [30].

Another factor influencing the decline in the registration of infections may be the significant decrease in demand for pediatric care. Numerous authors [28,33,34] have expressed their concern due to a noted hesitancy in the seeking of healthcare assistance by parents and caregivers, which was most likely due to a fear of exposure to coronavirus and limited access to primary care services, potentially delaying the diagnosis of other illnesses.

In addition to the reduced number of respiratory and other viral infections, this study demonstrated a major global decline in the prescription of antibiotics in primary care. This decrease continued throughout the study period and increased during the second quarter of 2020, coinciding with the at-home lockdown and stricter containment measures implemented during the pandemic. The most affected antibiotics are those typically used to treat respiratory infections, especially broad-spectrum penicillins, mainly amoxicillin, and combinations of penicillins and beta-lactamase inhibitors. It is interesting to note that the prescription of macrolides, especially azithromycin, despite being a treatment used in SARS-CoV-2 infections during the initial phase of the pandemic, revealed a significant reduction in use. Numerous studies have revealed that many COVID-19 patients received antibiotics, especially early on in the pandemic, and that these were mainly adults with comorbidities (75%). Children received lower rates (57%) [18,23,35,36]. 

Most bacterial infections occurring during pediatric ages can be treated in an outpatient manner through the prescription of oral antibiotics. These prescriptions tend to be written in primary and urgent care services, mainly for respiratory infections. This is the main cause of the inappropriate use of antibiotics [18,23,35,36,37]. In Spain, the age group that consumes the most antibiotics is that of 0 to 4 years, with an exposure rate of 58.8%, generally associated with inappropriate indication. The most commonly used antibiotics are amoxicillin and amoxicillin with clavulanic acid, with a tendency to increase the use of amoxicillin on its own. The second most commonly prescribed antibiotics are first- and second-generation macrolides and cephalosporins, with both of these experiencing a major decline in the rate of their prescription [32].

### 3.3. Impact on the Dispensing of Antibiotics

Data from this study on the consumption of antibiotics reveal that those from the J1C group (penicillins) are the most frequently used in all of the study years, followed by macrolides. This coincides with prior research [38].

A significant decline was not seen in the prescription of other antibiotics that are commonly used in non-respiratory infections (i.e., urinary tract infections), such as third-generation cotrimoxazole or cephalosporins, such as cefixime.

On the other hand, a minor increase was found in the use of tetracyclines, despite the fact that their use is quite limited in pediatrics, given their contraindication in children under the age of 8, their secondary effects, and the infrequency of the pathologies for which they are used.

Our data were compared with those published by the Spanish Agency of Medicines, on a national level (PRAN) [7], to verify whether or not there was support for the hypothesis regarding the declined use of antibiotics in 2020, secondary to the decrease in respiratory infections, as occurred in the Vigo health area. The data collected on a national level include adult and pediatric patients, as well as prescriptions from hospitals and private healthcare services. Therefore, the figures in our context are lower for all groups. The decrease in the use of antibiotics is clear for all groups, both on a national and local level, except for the J01E (trimethoprim and combinations) and J01X groups, which did not vary. This is consistent with the stability of the incidence of urinary infections and subsidiary pathologies treated with these therapeutic groups.

Divergence is evident in the use of antibiotics from group J01A (tetracyclines). The increased use of tetracyclines in the age group of 10 to 14 has been seen in our health area, but has not been observed on a national level. This may be related to the mandatory use of face masks by children over the age of 6, resulting in an increased number of cases of acne (the so-called maskne) [39]. The term maskne is used to refer to the acne resulting from the use of face masks by patients who did not previously suffer from this condition, since an area of heat, humidity, and friction is created, occluding the hair follicles and favoring the appearance of acne. This may have resulted in an increase in the prescription of tetracyclines for its treatment.

It is also necessary to consider the influence of telephone consultations in the prescription profiles, as compared to the face-to-face consultations. Studies comparing telephone care and in-person visits indicate that the former increases the number of follow-up consultations and the inappropriate prescription of antibiotics; however, additional studies are necessary in order to confirm the influence of telemedicine on this these types of prescriptions [40,41,42].

This decreased use of antibiotics has also been found in 25 other European countries [43], with the exception of Bulgaria, in which an increase took place in 2020. As noted in our study, the largest decrease took place in the use of penicillins and beta-lactams, drugs prescribed for respiratory infections, which is in line with the decline in the rate of these infections in the European Union (European Centre for Disease Prevention and Control) [44]. The theory of the ecological niche of viruses, according to which a dynamic variation of the ecosystem would be produced by being replaced by other viruses through the phenomena of competition or cooperation between them, would also have affected the circulation of SARS-CoV-2 in the year 2020, and would have caused the replacement of infections by other respiratory viruses [45].

In the infant and youth populations, there was a decreased circulation of viruses and bacteria as a result of the SARS-CoV-2 pandemic. There was also a decline in the consumption of antibiotics and a decrease in flare-ups of asthma in 2020 [27]. On the other hand, a significant increase was seen in psychosocial issues, such as anxiety, self-destructive behaviors, eating disorders, substance abuse, and new technology addictions. It should also be noted that restriction measures may cause increased susceptibility to other infective agents, which may result in more intense future or unexpected outbreaks [30]. Therefore, specific studies should assess which measures are more appropriate in a similar situation and which should be continued from the current practices (such as handwashing, staying at home when having symptoms of illness, and following clinical practice guidelines to ensure the rational use of antibiotics).

### 3.4. Limitations and Strengths

All of the primary care professionals used the same electronic clinical history and its use is universal in all visits. Furthermore, it may be considered representative of the infective episodes and the consumption of J01, given that 93.31% of the population has a digital clinical history on the National Healthcare System.

The main limitation of this study is the variability in the coding used by professionals and the lack of individual validation of the study events. However, we can assess its integrity and validity by comparing the rates obtained with those of the medical literature, or from their integration in the computerized system of primary care and public health. Thus, in the case of diseases of mandatory declaration (EDO, based on their initials in Spanish), in Galicia, the inclusion of episodes that have been recorded in the primary care clinical history of the epidemiological surveillance system of the EDO is standardized.

The data used in this study correspond to the outpatient use of J01 medications, not considering the consumption derived from hospital use, in admitted patients, or those dispensed to outpatients and in socio-healthcare establishments from hospital pharmacy services. It also excludes consumption via private prescriptions and mutual insurance agencies that is not invoiced through the SNS. It should also be noted that invoicing data do not correspond precisely with the data on the prescriptions, since said prescriptions may not be dispensed. Furthermore, not all medications dispensed by the pharmacies are actually taken by the patients.

ITS are increasingly recognized as designs that, when properly designed and analyzed, can provide reliable effect estimates on the impact of interventions in situations where randomization is not feasible [24]. Despite recent publications containing detailed guidance on how to conduct ITS studies [46], reporting on the design features and results of ITS studies is still lacking [47]. Among other noteworthy aspects, the specification of the time interval (months), the outcome (ratio or DHD), shape model (level change and slope change), autocorrelation application, and effect measures (level change and slope change, including *p* and confidence intervals) are specified. It was not possible to analyze a control group, as recommended in the literature. This was not available anywhere, on a global level.

## 4. Materials and Methods

### 4.1. Design and Study Population

This study was a real-world, observational, longitudinal, and retrospective data study on infectious episodes with ICPC-2 and oral medication in the J01 therapeutic subgroup, corresponding to anti-infective drugs for systemic use, according to the WHO’s ATC classification.

Administrative records from 2018 to 2020 for a primary care pediatric population in the Atlantic European health region were used. The region of Vigo (Galicia, Spain) was considered. It has a population of 569,534 inhabitants who are part of 43 basic health areas, with a healthcare center in each of these areas that has an average of 14,000 inhabitants per healthcare center, while the pediatric population comprises 13% of the total population.

In Spain, healthcare is based on a publicly financed system. Pediatric care is provided mainly by pediatricians, working with an assigned population in multi-professional healthcare centers.

### 4.2. Data Sources

For the episodes, the electronic clinical history from primary care was used. Clinical data were registered in the primary care system’s electronic health record (EHR) for at least the past ten years. Episodes were coded with ICPC-2. ICPC-2 contains 17 chapters, differentiated by one letter, corresponding to a mnemonic code in English. A list of codes identified as infectious and included in this project is presented in Appendix A.

For the medication, electronic prescriptions from the Galician Health Service (Servizo Galego de Saúde) were used, noting the active ingredients prescribed in primary care and dispensed by pharmacies. Invoicing data provided by the Galician pharmacies to the National Healthcare System (SNS) were obtained from the Subdirección Xeral de Farmacia do Servizo Galego de Saúde. When filling each prescription, the Personal Identification Code of the patient receiving the prescription, as printed on the healthcare card, was collected using the computer and was associated with the National Code of the medication during the prescription filling procedure. In this way, a file was generated, including the following variables: pharmaceutical specialty, number of packages, retail price, reference price, contribution (active/pensioner), expense to the Galician Healthcare Service, year, and month. 

For the study population, information from the Municipal Census of the National Statistics Institute was used [25].

### 4.3. Variables

For the episodes, the chapter and episode code, date of entry of the episode in the EHR (month and year), and five-year bracket age group, calculating the gross rate, were used.

For the medication, the regular daily dose (DHD) per active ingredient, month, and year was used. The DHD indicates the proportion of the population receiving standard treatment on a daily basis. It is calculated using the following formula:$$\mathrm{DHD}=\frac{\mathrm{UV}\times \mathrm{FF}/E\times C/\mathrm{FF}\times 1000}{\mathrm{DDD}\times \mathrm{No}.\mathrm{inhabitants}\times 365\mathrm{days}}$$

UV = units consumed. FF/E = no. of pharmaceutical forms per package. C/FF = quantity of active ingredients in each pharmaceutical form [23].

For the denominators, data from the pediatric population of the municipalities of Pontevedra were used, included in the Vigo area at the mid-way point of the period.

### 4.4. Statistical Analysis

A descriptive analysis of the episodes was performed by age, group, and ICPC group, as well as DHD by therapeutic subgroup with 4 digits.

Regarding the analysis of the time series, denoting their value by $X_{t}$ at time t, we considered the regression model $X_{t}=m\left(t\right)+e_{t}$, where the function m(t) represents the trend of the series and $e_{t}$ is the corresponding temporary error. In particular, in this article, we used the ARIMA or Box–Jenkins methodology as a statistical technique, which allows for mathematical models that establish the time structure (trend, seasonality, etc.) of the series. This type of model is very useful for short-term prediction.

On the other hand, we considered that the trend m(t) could be adjusted linearly in each *k* consecutive time period, $I_{1}=\left(t_{0},t_{1}\right],I_{2}=\left(t_{1},t_{2}\right],\ldots ,I_{K}=\left(t_{K-1},t_{K}\right]$, with the function (*K*) and position $\left(t_{0},t_{1},\ldots ,t_{K}\right)$, and in such a way that the trend m(t) is linear in each of these intervals. That is, $m\left(t\right)=a_{k}+b_{k}t$ if $t\in I_{k}$ for k=1,…,K.

In this paper, a bootstrap resampling plan was used to determine the number of *K* change points. In addition, an algorithm was proposed that, once the value of *K* is established, allows for the estimation of ${\hat{t}}_{0},\ldots ,{\hat{t}}_{K}$ of said points, and for the estimated coefficients ${\hat{a}}_{k}$ and $b_{k}$, corresponding to each of the *K* time intervals, to be obtained. Bootstrap resampling techniques can involve a high computational cost, increased in our case by a very large amount of data. For this reason, we used binning-type computational acceleration techniques. In this way, the results of the models can be obtained in a relatively short period of time.

Statistical analysis was carried out using IBM SPSS Statistic 21 and R-4.1.2.

### 4.5. Ethical Considerations

The request and handling of aggregate data guarantee the confidentiality of the same, given that no references are made to personal data, only to aggregate data by health area. The information has been provided by the Vigo Health Area (Servizo Galego de Saúde).

## 5. Conclusions

In this study, a decrease was observed in infectious/contagious diseases transmitted between individuals through the air and through the fecal–oral route, and a slight decrease was observed in infections transmitted via other mechanisms (such as urinary tract infections). In parallel, during these months, a notable decline was found in the global prescription of antibiotics. The analysis performed using segmented regression demonstrated the significant effect of the pandemic on both time series, which was higher than the prior seasonal variations experienced.

Specific studies should attempt to determine which non-pharmacological measures are more successful at containing the transmission of infectious illnesses. This may prevent the inappropriate prescription of antibiotics, which is fundamental for preventing future resistance and essential to improving the sustainability of the healthcare system, as well as for reducing costs and iatrogenesis.

## Figures and Tables

**Figure 1 antibiotics-11-00264-f001:**
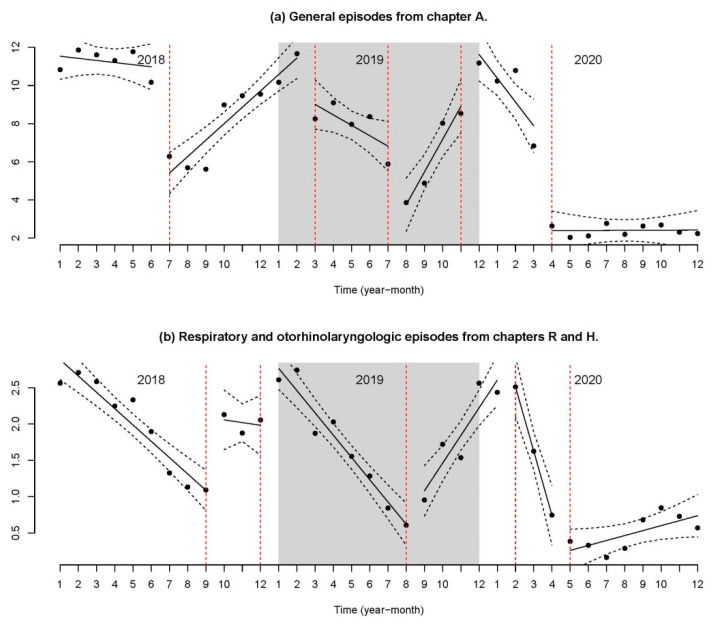
Segmented regression of the number of aggregated episodes. (**a**) General episodes from Chapter A. (**b**) Respiratory and otorhinolaryngological episodes from Chapters R and H.

**Figure 2 antibiotics-11-00264-f002:**
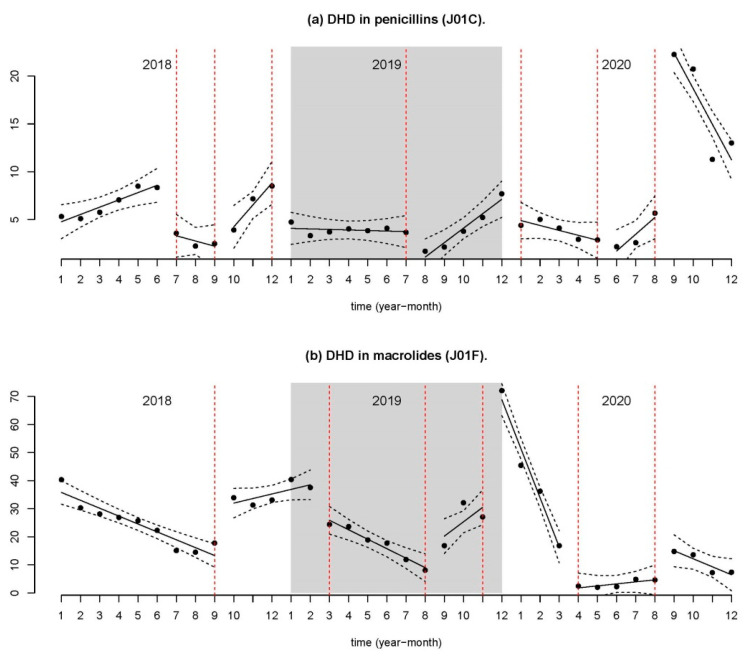
Segmented regression of the number of DHD. (**a**) DHD in penicillins (J01C). (**b**) DHD in macrolides (J01F).

**Figure 3 antibiotics-11-00264-f003:**
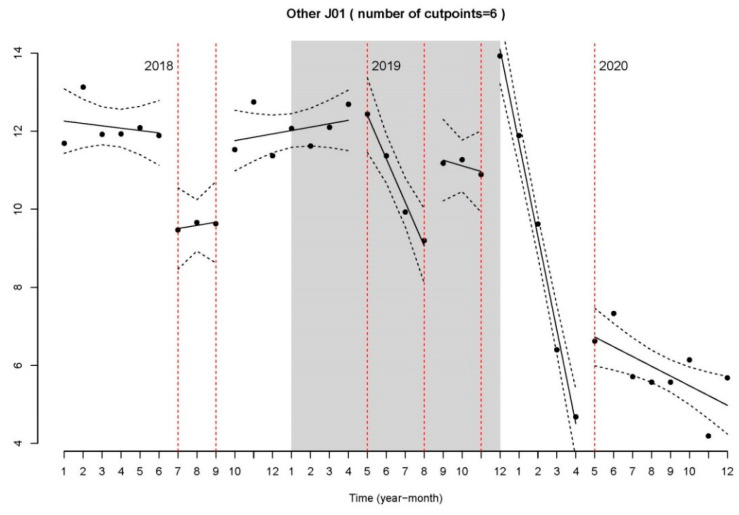
Segmented regression of the number of DHD in other antibiotics.

**Table 1 antibiotics-11-00264-t001:** Number of episodes by ICPC-2 chapter, five-year age group, and year.

	00–04			05–09			10–14		
ICPC-2 Chapters	2018	2019	2020	2018	2019	2020	2018	2019	2020
A General and Unspecified	3243	2171	2793	1407	908	2428	401	425	1807
B Blood	3			4	5	2	1	2	2
D Digestive	1501	1502	476	731	712	302	639	651	297
F Eye	2011	1874	852	677	641	398	468	409	274
H Ear	3196	3242	1227	1265	1070	608	877	681	536
K Cardiovascular		1		1		1	1	1	
L Musculoskeletal	1	2	1	1	3	2	1	3	
N Neurological	8	27	3	7	7	3	2		1
R Respiratory	11,330	12,927	4956	4095	3730	2174	3022	2622	1932
S Skin	527	645	455	1264	1282	904	1035	1061	780
U Urological	400	338	311	280	250	206	165	137	111
X Female genital	26	24	17	7	3	4	20	24	16
Y Male genital	241	291	195	128	114	108	51	44	35
Total	22,487	23,044	11,286	9867	8725	7140	6683	6060	5791

**Table 2 antibiotics-11-00264-t002:** Number of DHD by therapeutic subgroup and year.

	Year		
N. DHD by Therapeutic Subgroup	2018	2019	2020
J01AA Tetracyclines	0.18	0.27	0.37
J01CA Penicillins with extended spectrum	5.17	4.79	2.13
J01CE Beta-lactamase-sensitive penicillins	0.13	0.11	0.05
J01CF Beta-lactamase-resistant penicillins	0.01	0.00	0.00
J01CR Combinations of penicillins, including beta-lactamase inhibitors	4.10	3.23	1.90
J01DB 1st-generation cephalosporins	0.01	0.01	0.00
J01DC 2nd-generation cephalosporins	0.56	0.55	0.31
J01DD 3rd-generation cephalosporins	0.16	0.16	0.14
J01EA Trimethoprim and derivatives	0.00	0.00	0.00
J01EE Combinations of sulfonamides and trimethoprim	0.06	0.06	0.05
J01FA Macrolides	1.99	1.68	0.93
J01FF Lincosamides	0.01	0.00	0.00
J01MA Fluoroquinolones	0.01	0.00	0.00
J01XC Steroid antibiotics	0.00	0.00	0.00
J01XE Nitrofuran derivatives	0.00	0.00	0.00
J01XX Other antibiotics	0.03	0.03	0.03

## Data Availability

Requests for any underlying data cannot be granted by the authors because the data were acquired according to a Galician Health Service internal procedure; therefore, conditions of use (and further use) apply.

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
