# Peer review of "Interrupted Time Series Analysis of Pediatric Infectious Diseases and the Consumption of Antibiotics in an Atlantic European Region during the SARS-CoV-2 Pandemic"

_antibiotics, 2022, doi:10.3390/antibiotics11020264_

Round 1

Reviewer 1 Report

Dear Authors; the paper idea is novel and interesting. However, it has some "very serious" issues listed below. Regards,

P.S.

[1] Writing:

1-1 The abstract of the paper has serious "focus" issue. Please rewrite it. (i)Background, (ii) Objectives, (iii) Methods, (iv) Results, (v)Conclusions, (vi) implications. Each of these subcomponents should 2-3 sentences.

1-2 Line 109-110: needs reference , line 117: needs more references

1-3 Reorder the sections of the article: 1-Introduction, 2- Materials&Methods, 3- Results, 4-Discussion, 5-Conclusions

1-4 Figure.1. & Figure.2:  Add panel (a) to the top and panel (b) to the bottom sub-figure. Same in the captions

1-5 Add list of Abbreviations used in the paper before the reference section for the readers referral. Example:

Abbreviation

ITS  Interrupted Time Series

1-6 Table2: The items are in Spanish.   They should be in English. 

[2] Statistical: 

2-1 appendix B: Please remove all the R outputs and instead write tables in latex with captions(the current ones are subprofessional in medical statistical writing).

2-2 Line 409 & 418: Merge the two in one line time series formula with number

2-3 Line 399: Number the equation. Type it correctly.

2-4 Line 425:  Write the name of Statistical software used in the paper. I think it is "R". Add one citation to R statistical software. Name which packages and libraries you used in the paper. 

2-5 In the paper the consumption level of three antibiotics are mentioned as the outcome of interest. Couple questions:

2-5-1. Did you check potential "multicolinearity" of these three outcomes ? the current statistical analysis makes sense when there is no multicolinearity between these three outcomes. Please add the results of multicolinearity analysis to this paper in one paragraph.

2-5-2. It is highly recommended in one extra subsection in the results section, the authors add the summation of consumption levels of the three mentioned antibiotics called "overall antibiotic consumption"  in 2-5-1 and plot the segmented regression Figure and comment on it.  

Reviewer 2 Report

Ana Claveria et al. used interrupted time series analysis to analyze infectious diseases and antibiotics prescription in pediatric population, from 2018 to 2021. The manuscript is well written, clear and organized, and the results and discussions are well presented and scientifically sound. This paper would be ready for publication after some minor changes outlined below:

  1. On Table 1, some numbers are missing the last digit. For example, R-respiratory, 00-04, 2018 has the number 11,33. It seems to be missing the last digit. You can also find similar error for H ear (1,07), and Total (7,14 and 6,06). Please make appropriate corrections for them
  2. For Table 2, please replace the Spanish words with English (N. DHD por subgrupo terapéutico) and (Año)
  3. On line 174, the authors mention that 'In parallel, a significant increase is evident in the prescribing of antibiotics, especially penicillins, cephalosporins, and macrolides." However, their prescriptions decrease, not increase, in 2020. Please make appropriate adjustment if this is an error in writing, or explain by pointing to data if the author really mean that there is significant increase
  4. On line 258, the authors noted that "prescription of macrolides, especially azithromycin, despite being commonly used to treat respiratory infections, did not reveal a significant reduction in use". However, when I refer to Figure 2 (J10F), it looks like DHD for macrolides decreases from 2.5 to around 1, so 250% decrease. That is quite a lot. Can the author define what they mean by significant? And clarify why this 250% decrease do not count as significant reduction in use?

Author Response

 (REVIEWER 2)

COMMENT 1. On Table 1, some numbers are missing the last digit. For example, R-respiratory, 00-04, 2018 has the number 11,33. It seems to be missing the last digit. You can also find similar error for H ear (1,07), and Total (7,14 and 6,06). Please make appropriate corrections for them.

Corrections made. Thanks.

COMMENT 2. For Table 2, please replace the Spanish words with English (N. DHD por subgrupo terapéutico) and (Año).

Thanks for pointing this out, we have translated Spanish words into English, as it is shown before in comment 1.6.

COMMENT 3. On line 174, the authors mention that 'In parallel, a significant increase is evident in the prescribing of antibiotics, especially penicillins, cephalosporins, and macrolides." However, their prescriptions decrease, not increase, in 2020. Please make appropriate adjustment if this is an error in writing, or explain by pointing to data if the author really mean that there is significant increase.

In response to this comment, it is indeed an error in writing.

COMMENT 4. On line 258, the authors noted that "prescription of macrolides, especially azithromycin, despite being commonly used to treat respiratory infections, did not reveal a significant reduction in use". However, when I refer to Figure 2 (J10F), it looks like DHD for macrolides decreases from 2.5 to around 1, so 250% decrease. That is quite a lot. Can the author define what they mean by significant? And clarify why this 250% decrease do not count as significant reduction in use?

We appreciate the reviewer's observation and we have changed the writing to make it more clear.

“[…] It is interesting to note that the prescription of macrolides, especially azithromycin, despite being a treatment used in SARS-CoV-2 infections during the initial phase of the pandemic, revealed a significant reduction in use. Numerous studies have revealed that many COVID patients received antibiotics, especially early on in the pandemic, mainly adults with comorbidities (75%). Children revealed lower rates (57%) [34-37] “. 

Reviewer 3 Report

This is a very interesting interrupted time series study of the impact of the COVID pandemic on infectious diseases incidence and antibacterial agent utilization in children residing in Vigo, Galicia, Spain. The manuscript is well-written and cited. I have only several comments that would strengthen an already solid paper.

Title: very descriptive of the study

Abstract:

at line 33, please change "We intend to" to " The effect of the SARS-CoV-2 pandemic on the incidence of infectious diseases and the use of antibiotics was quantified and analysed."

At line 38, spell out ICPC-2 and add World Health Organization (WHO) to Anatomical Therapeutic Chemical Classification (ATC). At line 41, change 'a significant effect' to ' a significant decrease'.

At line 44, change 'a significant reduction in the antibiotics' to 'a significant reduction in anti-bacterial agent utilisation.'

Key words: change 'antibiotic consumption' to 'anti-bacterial agents' and add 'drug utilization' and 'Spain'. All other terms are appropriate.

Introduction: excellent description of the context in this region.

Results: well presented. Table 2 is in Spanish. It's understandable, and to be consistent with the manuscript, probably should be presented in English.

Methods: well described and documented with appropriate data source utilization.

Discussion: limitations are well outlined and future research direction is insightful.

References: not in mdpi style.

Thank you for the opportunity to review and comment on this important manuscript.

Author Response

COMMENT 1. Please Change in the abstract:

Following the reviewer's suggestion, we have changed along the entire abstract. The may not be in the same line because we have rewritten and restructured the abstract.

1.1 At line 33, please change "We intend to" to " The effect of the SARS-CoV-2 pandemic on the incidence of infectious diseases and the use of antibiotics was quantified and analysed."

“[…] The effect of the SARS-CoV-2 pandemic on the incidence of infectious diseases and the use of antibiotics was quantified and analysed […] “

1.2 At line 38, spell out ICPC-2 and add World Health Organization (WHO) to Anatomical Therapeutic Chemical Classification (ATC).

Changed to:

according to World Health Organization Anatomical Therapeutic Chemical (ATC) Classification System”

1.3. At line 41, change 'a significant effect' to ' a significant decrease'.

Changed to:

“[…]. A significant decrease during the SARS-CoV-2 pandemic is demonstrated in infectious diseases […]”

1.4. At line 44, change 'a significant reduction in the antibiotics' to 'a significant reduction in anti-bacterial agent utilization.'

“[…] there has been a marked a significant reduction in anti-bacterial agent utilization […]”

1.5. Key words: change 'antibiotic consumption' to 'anti-bacterial agents' and add 'drug utilization' and 'Spain'. All other terms are appropriate.

“Keywords: Interrupted time series analysis; pediatric; infectious diseases; SARS-CoV-2; anti-bacterial agents; drug utilization; Spain.”

COMMENT 2: References: not in mdpi style.

We appreciate the reviewer's observation and we have put them in mdpi style.

Round 2

Reviewer 1 Report

Dear Authors; most of my concerns were addressed satisfactorily. Regards.

Author Response

Thanks.

Your comments were very suggestive. We will continue working on segmented regression, and they will help us.